# Minimally Invasive Isolated Aortic Valve Replacement in a Potential TAVI Cohort of Patients Aged ≥ 75 Years: A Propensity-Matched Analysis

**DOI:** 10.3390/jcm12154963

**Published:** 2023-07-28

**Authors:** Ali Taghizadeh-Waghefi, Asen Petrov, Philipp Jatzke, Manuel Wilbring, Utz Kappert, Klaus Matschke, Konstantin Alexiou, Sebastian Arzt

**Affiliations:** 1Medical Faculty “Carl Gustav Carus”, Technical University of Dresden, 01307 Dresden, Germany; 2Center of Minimally Invasive Cardiac Surgery, University Heart Center Dresden, Faculty of Medicine, Technical University of Dresden, 01037 Dresden, Germany; 3Anesthesiology and Intensive Care Medicine, Dresden University Hospital, Faculty of Medicine, Technical University of Dresden, 01307 Dresden, Germany

**Keywords:** surgical aortic valve replacement, minimally invasive aortic valve replacement, minimally invasive surgery, outcomes, TAVI

## Abstract

(1) Background and Objectives: Transcatheter aortic valve implantation is guideline-recommended from the age of 75. However, this European guideline recommendation is based on limited evidence, since no interaction between age and primary outcome has been found in guideline-stated references. This study aimed to compare the short-term outcomes of minimally invasive isolated aortic valve replacement in patients aged ≥ 75 with those of younger patients; (2) Patients and Methods: This retrospective cohort study included 1339 patients who underwent minimally invasive isolated aortic valve replacement at our facility between 2014 and 2022. This cohort was divided into two age-based groups: <75 and ≥75 years. Operative morbidity and mortality were compared between groups. Further analysis was performed using propensity score matching; (3) Results: After matching, 347 pairs of patients were included and analyzed. Despite the higher EuroSCORE II in the ≥75 group (2.2 ± 1.3% vs. 1.80 ± 1.34%, *p* ≤ 0.001), the 30-day mortality (1.4% vs. 1.2%; *p* = 0.90) and major adverse cardiac and cerebrovascular events, such as perioperative myocardial infarction (0.0% vs. 1.2%, *p* = 0.12) and stroke (1.4% vs. 2.6%, *p* = 0.06), were comparable between both treatment groups; (4) Conclusions: Minimally invasive aortic valve replacement is a safe treatment method for patients aged ≥ 75. Our results indicate that the unilateral cut-off of 75 years is not a limiting factor for performing minimally invasive aortic valve replacement.

## 1. Introduction

The 2021 European Society of Cardiology and European Association for Cardio-Thoracic Surgery guidelines for the treatment of valvular heart disease recommend an age cut-off of ≥75 years as a decision-making criterion for selecting therapeutic procedures for severe aortic stenosis (AS) in favor of transcatheter aortic valve implantation (TAVI) [1]. The Class I recommendation with level of evidence grade A is notable in this regard. Previous clinical studies that form the basis of this recommendation relate to the PARTNER [2,3,4], SURTAVI [5], Corevalve high-risk trial [6] and NOTION [7] trials. However, except for the NOTION trial, where patients aged ≥ 70 years were randomly assigned to surgical aortic valve replacement (SAVR) or TAVI, these studies did not primarily evaluate age-based outcomes. Therefore, this recommendation on patient age has a high risk of methodological error. Furthermore, an evidence-based answer to whether patients ≥ 75 years derive a clear benefit from the TAVI procedure remains elusive. Not to be neglected are the progressive advances in surgical techniques for isolated aortic valve replacement, which eventually led to the development of minimally invasive aortic valve replacement (MIAVR). MIAVR is increasingly being performed as an alternative to standard sternotomy to meet the rising patient demand for faster postoperative recovery and improved quality of life. Therefore, this study reports the short-term postoperative results and outcomes of MIAVR in patients aged ≥ 75 with a life expectancy of >5 years compared to younger patients.

## 2. Materials and Methods

### 2.1. Patient Population, Study Design and Ethics Statement

This study is a retrospective observational cohort analysis of consecutive patients undergoing minimally invasive aortic valve surgery according to the inclusion criteria. Data were collected from the hospital database. This study was reviewed and approved by the local Ethics Board (EK—Nr. 298092012).

Generally, all patients being admitted with a symptomatic high-grade aortic valve stenosis undergo a Heart Team decision process. During this decision-making process, patients who are likely to have a life expectancy of ≤5 years due to their age or comorbidities are usually assigned to a TAVI procedure. The remaining patients are primarily screened for MIAVR. This prospective decision making is represented in the retrospectively analyzed cohort, according to the flow chart depicted below (Figure 1). Sternotomy patients were ruled out to exclude any selection bias.

Between 2014 and 2022, 2171 patients underwent elective isolated primary SAVR at our facility due to degenerative aortic valve disease. Eight hundred thirty-two patients who underwent full sternotomy were excluded. Therefore, this retrospective analysis included 1339 patients who underwent isolated MIAVR. This cohort was divided into two age-based groups: <75 years and ≥75 years. Demographic, clinical, surgical and short-term postoperative outcome data were obtained from medical records retrospectively. Figure 1 shows the patient selection process as a flow chart.

### 2.2. Surgical MIAVR Access Routes

Three different access routes for MIAVR are performed at our facility (Figure 2). The choice of minimally invasive surgical approach was based on the surgeon’s preference and the patient’s anatomic characteristics, which were assessed using electrocardiogram (ECG)-gated computed tomography angiography of the thorax, abdomen and pelvis. The surgical access routes consisted of upper partial sternotomy (UPS), right anterolateral thoracotomy (RAT) and right lateral thoracotomy (RLA). In UPS, a 5–10 cm median skin incision is made below the sternal notch, followed by a J-shaped division of the upper part of the sternum up to the level of the third or fourth intercostal space. RAT involves a 5 cm skin incision and dissection of the pectoralis and intercostal muscles and the right mammary artery along the second intercostal space. RLA uses a 5 cm skin incision made along the right anterior axillary line, followed by dissection of serratus anterior and intercostal muscle and opening of the third or fourth intercostal space [8]. Regardless of the surgical access route, extra-corporal circulation was established via the femoral vessels, placing antegrade cardioplegia cannula in the ascending aorta and the left ventricular venting line via the right superior pulmonary vein.

### 2.3. Statistical Analysis

Continuous data are expressed as mean ± standard deviation (SD). Categorial variables are presented as numbers and percentages. All continuous variable data were checked for normality using the Kolmogorov–Smirnov test with Lilliefors significance correction (type I error = 10%). Normally distributed variables were tested for variance heteroscedasticity using Levene’s test (type I error = 5%). Normally distributed variables with homogeneous variance were compared between subgroups using independent two-sample *t*-tests. Normally distributed variables with heterogeneous variance were compared using Welch’s *t*-test. Non-normally distributed and ordinal variables were compared using Mann–Whitney’s U-test. Dichotomous variables were compared using Fisher’s exact test. Other categorical variables were compared using the χ^2^ test (either exact or with Monte Carlo simulation). For individual categories, deviations from the expected frequencies are presented as adjusted residuals. Since the type I error was not adjusted for multiple testing, the results of inferential statistics are descriptive only, and the use of the term “significant” in the description of the study results always reflects only a local *p* < 0.05 but no error probability below 5%.

A bias-reduced subset of the full data set was obtained by propensity score matching of the following variables: sex, body mass index (BMI), preoperative left ventricular ejection fraction (LVEF), estimated creatinine clearance according to the Cockcroft–Gault equation (CRCL), preoperative New York Heart Association (NYHA) classification, diabetes mellitus, pulmonary arterial hypertension, coronary artery disease, peripheral occlusive arterial disease and chronic obstructive pulmonary disease (COPD). A maximum allowable difference between two patients of 0.08 was defined to ensure good matches. The choice of the caliper value of 0.08 was based on ensuring that the two groups being compared are as similar as possible while also maximizing the number of matches. It was started with a higher caliper value, which was gradually decreased until the balance between the similarity (no statistically relevant differences between the propensity-score-matching parameters regarding the groups) and the number of matches was satisfactory. Hence, the stated caliper value was used for the matching. The standardized differences are shown in Table 1. Figure 3 shows the covariate balance before and after adjustment. Statistical analyses were performed using the open-source R statistical software (v.4.1.2).

## 3. Results

### 3.1. Patient Baseline Characteristics

Between January 2014 and February 2022, 1339 patients underwent MIAVR and were divided into two age-based groups based on a cut-off of 75 years, with 956 patients in the isolated MIAVR < 75 group (mean age = 64.2 ± 8.1) and 383 patients in the isolated MIAVR ≥ 75 group (mean age = 77.3 ± 1.7). The sex distribution differed in the two groups, with more males in the <75 group (65.2%) than in the ≥75 group (56.9%; *p* ≤ 0.01). Patients in the ≥75 group had a significantly higher mean calculated risk stratification score for mortality (EuroSCORE II = 2.25 ± 1.31% and STS-PROM score = 1.91 ± 0.9%) than those in the <75 group (1.38 ± 1.03 with *p* ≤ 0.001 and 1.14 ± 0.67 with *p* ≤ 0.001, respectively). They were also more likely to be NYHA class III or IV and to have more comorbidities, such as arterial hypertension, diabetes mellitus, coronary artery disease, pulmonary arterial hypertension, renal insufficiency, significant carotid artery stenosis (stenosis ≥ 50% measured according to the NASCET criteria) and atrial fibrillation. There were significantly more smokers in the <75 group than in the ≥75 group.

We could propensity score match 347 patients from each group in the full cohort according to the above-mentioned variables, so that both groups had similar baseline characteristics. After the matching procedure, the main significant differences between the <75 and ≥75 groups were age (77.2 ± 1.7 vs. 67.8 ± 5.9; *p* ≤ 0.001), as expected, and under the influence of age, the calculated risk stratification scores for mortality (EuroSCORE II 2.2 ± 1.3% vs. 1.80 ± 1.34%, *p* ≤ 0.001; STS-PROM score 1.9 ± 0.9% vs. 1.5 ± 0.8%, *p* ≤ 0.001). Moreover, patients in the ≥75 group more frequently had arterial hypertension, significant carotid stenosis and implanted permanent pacemakers than patients in the <75 group. Patients with preoperative terminal-dialysis-dependent renal insufficiency were significantly more common in the <75 group than in the ≥75 group. Other baseline characteristics did not differ significantly after matching. The baseline characteristics of the pre-matched and propensity-matched groups are shown in Table 2.

### 3.2. Unadjusted Outcomes

Significantly more patients underwent MIAVR through the RAT surgical access route in the ≥75 group (42.0%) than in the <75 group (34.6%; *p* ≤ 0.05; Figure 4). There were no significant differences in the use of the other two surgical access routes between groups. As expected, mechanical valve prostheses were implanted more frequently in the <75 group than in the ≥75 group (*p* ≤ 0.05). However, significantly more rapid deployment bioprosthetic valves (RDV) were implanted in the ≥75 group than in the <75 group (*p* ≤ 0.05; Figure 5). Among the procedural data, there was a significantly longer cardiopulmonary bypass time (CPBT; 66.9 ± 21.8 vs. 64.4 ± 24.5; *p* = 0.02) and aortic cross-clamp time (ACCT; 46.8 ± 16.5 vs. 43.6 ± 14.8; *p* ≤ 0.001) in the <75 group than in the ≥75 group (Figure 6). The procedural and intraoperative data are summarized in Table 3 and Table 4.

The primary postoperative ventilation time was shorter in the <75 group than in the ≥75 group (*p* = 0.04). Transfusion of packed red blood cells (PRBC) was performed less frequently in the <75 group than in the ≥75 group. Postoperative acute kidney injury (AKI) occurred more frequently in the ≥75 group (9.9%) than in the <75 group (4.9%; *p* ≤ 0.001). When classified by grade, AKI grade II or III occurred more frequently in the ≥75 group (7.3%) than in the <75 group (3.8%; *p* = 0.01). Patients in the ≥75 group were significantly more likely to develop postoperative delirium than patients in the <75 group. Postoperative new-onset atrial fibrillation (NOAF) was significantly more common in the ≥75 group than in the <75 group. However, there were no significant differences in the 30-day mortality rate (1.5% vs. 1.8%; *p* = 0.82) or incidence of postoperative major adverse cardio-cerebral (MACCE) events, such as stroke (2.2% vs. 2.1%; *p* ≥ 0.99), transient ischemic attack (0.9% vs. 0.8%; *p* ≥ 0.99) and perioperative myocardial infarction (MI), between groups (Table 5).

The echocardiographic data obtained at discharge showed optimal prosthesis function in all patients. The transaortic peak and mean pressure were reduced significantly and did not differ between groups. Paravalvular regurgitation occurred at a similar frequency in both groups (Table 6).

### 3.3. Outcomes of Propensity-Score-Matched Patients

After propensity score matching, we obtained 694 patients (347 pairs) for the matched analysis. Consistent with the unadjusted analysis, mechanical valves were more frequently implanted in the propensity-matched <75 group than in the ≥75 group (Figure 5).

Other procedural and intraoperative variables did not differ significantly between propensity-matched groups. The detailed procedural and intraoperative data are shown in Table 2 and Table 3. The rates of postoperative delirium (26.9% vs. 19.1%; *p* = 0.02) and NOAF (18.8 vs. 11.3; *p* = 0.01) were significantly higher in the propensity-matched ≥75 group than in the <75 group. However, the 30-day mortality, MACCE and composite major morbidity did not differ significantly between groups (Table 5). Echocardiographic data did not differ significantly between groups (Table 6). The graphical overview of the postoperative outcomes of propensity-matched groups is shown in Figure 7.

## 4. Discussion

As the indication for catheter-based valve procedures has expanded from high-risk patients to increasingly intermediate- and low-risk patients, TAVI has fundamentally changed the treatment regimen for patients with aortic valve stenosis. This trend found its way into the 2021 ESC/EACTS guidelines, postulating an age threshold favoring TAVI in patients ≥75 years [1]. The 2020 ACC/AHA guidelines go even further, discussing an age threshold of 65 years [8].

Nevertheless, the results from the OBSERVANT trial and the GARY registry have recently shown that TAVI patients have worse survival and a higher risk of serious cardiac and cerebrovascular events at five years compared with SAVR [9,10]. Additionally, a recent meta-analysis by Barili and colleagues demonstrated that “TAVI becomes a risk-factor for all-cause mortality and the composite endpoint [of death or stroke] after 24 months and for rehospitalization after 6 months” [11]. However, the ESC/EACTS guidelines also contain an interesting statement—but only for the curious readers, whose interest goes beyond the tables included. It is said therein that an age cut-off might be problematic, and in the individual case, the patient’s life expectancy and the assumed durability of the catheter heart valve must be weighed up [1].

Considering this statement, taken together with the data published by Barili and colleagues—knowing that the average life expectancy of women and men aged 75 in Europe is 12–15 and 10–12 years, respectively—one could argue that every patient who has a life expectancy exceeding 5 years should undergo surgery, independent of their numeric age [11,12].

These considerations overshadow the unilateral support for the guideline recommendation in favor of TAVI for patients ≥ 75 years or even 65 years. Most previous randomized controlled clinical trials have compared SAVR with TAVI, usually focusing on the assessed surgical risk. Neither of these studies evaluated outcomes based on age, except for the NOTION trial. Furthermore, no distinction was made between conventional and minimally invasive access methods. We believe this is a methodical shortcoming. Recently, Wilbring et al., from our working group, demonstrated in a large-scale propensity-matched trial that transaxillary MIAVR was at least as safe as conventional SAVR using sternotomy but had the advantages of shorter hospital stay, shorter ventilation times, fewer transfusions, shorter ICU stay and bisected expected vs. observed mortality [13]. These results—with awareness of the increasing patient demand for less trauma, less pain and faster recovery—cast a slur on sticking to the classic sternotomy approach. It is quite understandable that in the age of TAVI, no patient is really convinced of sternotomy. MIAVR can be a strong argument in the discussion with a patient, as it is in Heart Team’s discussion. At our institution, we advocate an institutionalized MIAVR strategy. This resulted in a 97.2% MIAVR rate in isolated aortic valve surgery, abolishing sternotomy almost completely and helping increase the number of SAVR procedures by around 20% from 2014 to 2022 [14].

Therefore, this study only reports the outcomes of isolated MIAVR in 1339 patients divided into two age-based groups according to a 75-year cut-off, followed by a propensity-score-matching analysis of 694 patients (347 pairs). No statistically significant difference in 30-day mortality was detected with either unadjusted or propensity-score-matched data. However, the mortality risk stratification scores for 30-day mortality (EuroSCORE II and STS-PROM) were significantly higher in the ≥75 group than in the <75 group. Similarly, no significant differences were found in perioperative stroke, transient ischemic attack (TIA) or MI incidence. The surgical arm of the NOTION trial, with a mean EuroSCORE II of 2.0% and mean age of 79 years in the full cohort, reported a notably higher 30-day mortality rate (3.7%) and perioperative stroke incidence (3.0%) than our patients aged ≥ 75 years. It should be noted that the recording of perioperative stroke events in our study was completely independent of clinical symptom severity or graduation according to the modified Rankin scale. Therefore, stroke event was assessed in case of any neurological deficits, and the corresponding computed tomographic data were correlated.

Previous studies identified older age as an independent NOAF predictor after SAVR [15,16,17]. Consistent with the results of these studies, postoperative NOAF occurred significantly more frequently in patients ≥ 75 years of age in our study. Advanced age is also an independent risk factor for postoperative delirium after cardiac surgery [18]. In accordance with these findings, the incidence of postoperative delirium was higher in the ≥75 group than in the <75 group in both the pre-matched and propensity-matched analyses.

In the pre-matched analysis, longer CPBT and ACCT were observed in the <75 group than in the ≥75 group. The cause of this observation remains unclear in the present data, especially since it was not reproduced in the propensity-score-matched analysis. The need for blood transfusion was significantly higher in the ≥75 group than in the <75 group in the unadjusted analysis. One possible reason for this could be the higher incidence of anemia in older age [19]. However, this finding was not reproduced in the propensity-matched analysis. The higher AKI rate in the pre-matched ≥ 75 group than in the <75 group was also not reproduced in the propensity-matched cohort. The higher AKI rate in the unadjusted ≥75 group, most likely related to already decreased preoperative creatinine clearance, did not differ significantly between the propensity-matched groups.

## 5. Limitations

This study has inherent limitations. First and foremost, despite its large cohort, this was a single-center retrospective study with only short-term follow-up. Second, our propensity-score-matching model might not have incorporated unknown but potentially relevant risk factors and confounders. Another potential problem arises from the fact that matching parameters were selected primarily based on surgical feasibility in minimally invasive aortic valve replacement procedures, resulting in incomplete matching of baseline characteristics (arterial hypertension, carotid artery stenosis, preoperative permanent pacemaker and preoperative hemodialysis). Furthermore, these MIAVR results were achieved in a high-volume expert center and cannot be extrapolated to all patients.

## 6. Conclusions

The aim and perspective of the present study are founded in the belief that TAVI is a great therapy, which has profoundly changed the treatment of valvular heart disease. Nonetheless, the decision-making process for TAVI or SAVR must be based on evidence. While TAVI has gained recognition as an effective treatment for high-grade aortic valve stenosis, we question the evidence-based data supporting the recommended age limit of 75 years. The present European guidelines suggest that an age threshold of 75 years is a long-standing legitimate decision parameter, despite the lack of evidence for “75 years” [1,20,21]. In our study, we aimed to compare the short-term outcomes of MIAVR in two age groups, specifically those below and above 75 years, rather than comparisons with a cohort of patients who underwent TAVI. The focus was to investigate whether patients above 75 years of age experienced any disadvantages in terms of short-term outcomes compared to the younger group. Our study data clearly indicate that this was not the case. Therefore, in addition to presenting our study findings, we aim to provoke a reconsideration of this somewhat arbitrary age limit. In addition, this study may help improve the decision-making process for or against SAVR.

A further aspect is that it was shown by means of hard endpoints, such as survival and stroke, that MIAVR is at least not inferior to sternotomy. Furthermore, minimally invasive techniques unarguably find wider patient acceptance because they meet the demand for less trauma, less pain and better cosmesis. In counterpoint to TAVI, it is imperative to question the following thesis: given the well-documented advantages offered by modern minimally invasive techniques for aortic valve replacement compared with sternotomy, what justifies the hesitation to firmly establish minimally invasive aortic valve replacement (MIAVR) as the prevailing standard?

Overall, we hope to encourage the advancement of cardiac surgery and the development of minimally invasive surgical approaches as suitable therapeutic options for older patients. Although our study’s observation period was short, the results suggest that the minimally invasive approach is justified for patients above 75 years of age as well.

## Figures and Tables

**Figure 1 jcm-12-04963-f001:**
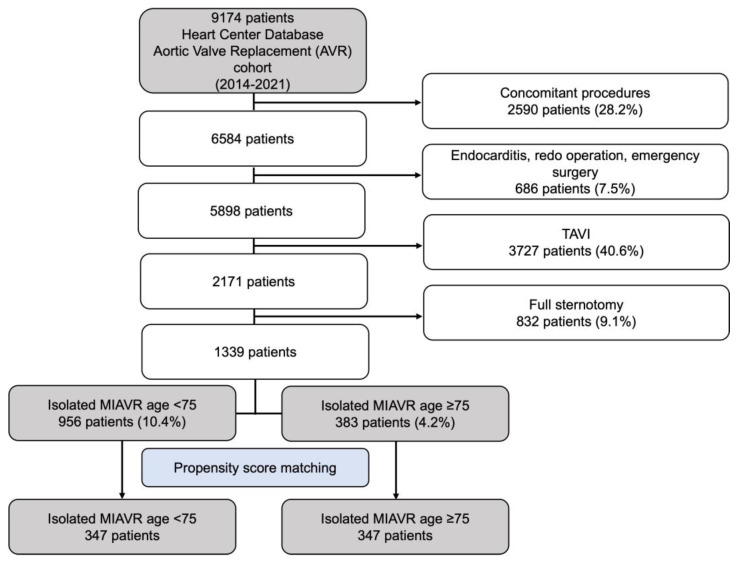
Flow diagram of the study population. Abbreviations: MIAVR, minimally invasive aortic valve replacement.

**Figure 2 jcm-12-04963-f002:**
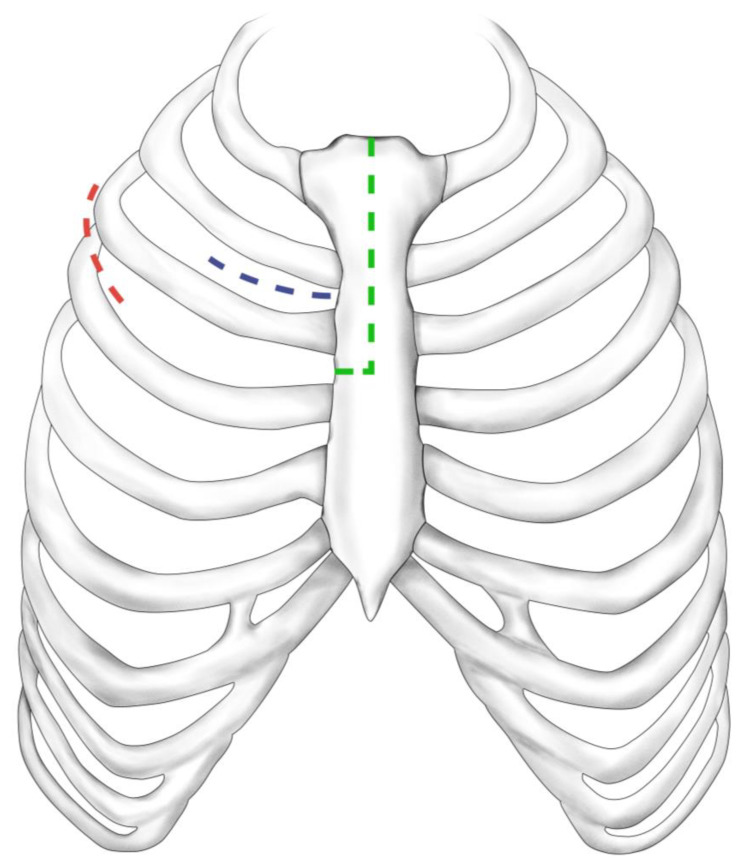
Surgical access routes. Upper partial sternotomy (green dashed line), right anterolateral thoracotomy (blue dashed line), right lateral thoracotomy (red dashed line).

**Figure 3 jcm-12-04963-f003:**
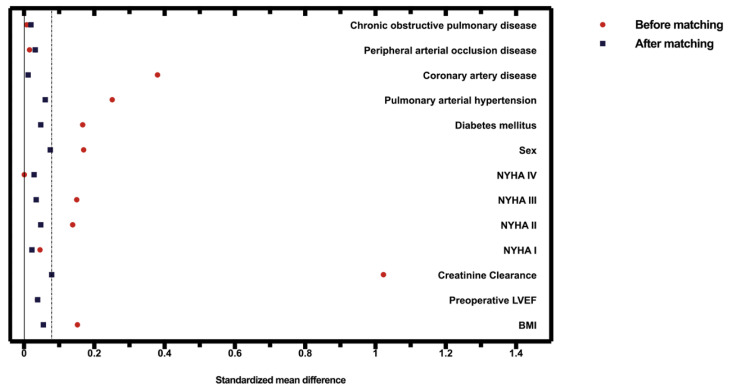
The covariate balance before and after propensity score matching. The dashed line represents the matching caliper of 0.08.

**Figure 4 jcm-12-04963-f004:**
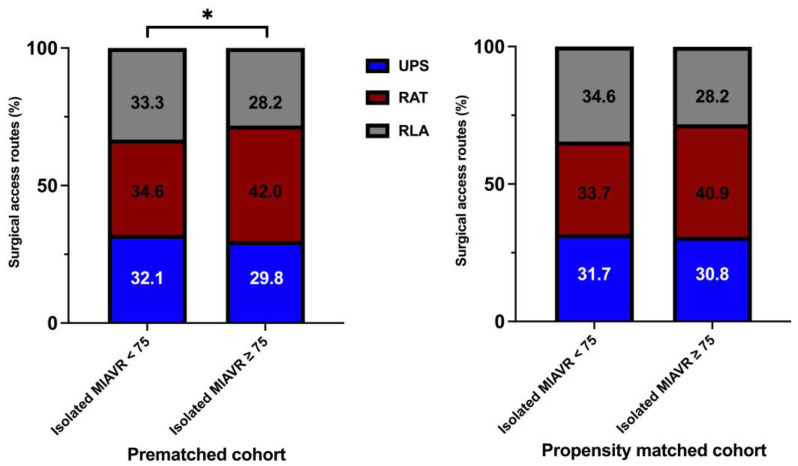
Distribution of surgical access routes within each treatment group. Abbreviations: UPS, upper partial sternotomy; RAT, right anterolateral thoracotomy; RLA, right lateral thoracotomy. Note: * *p* < 0.05 between groups. Note: As a consequence of the process of mathematical rounding, wherein percentages are rounded up or down to the nearest tenth decimal place, minute deviations of up to 0.1% from the absolute value of 100% can potentially manifest.

**Figure 5 jcm-12-04963-f005:**
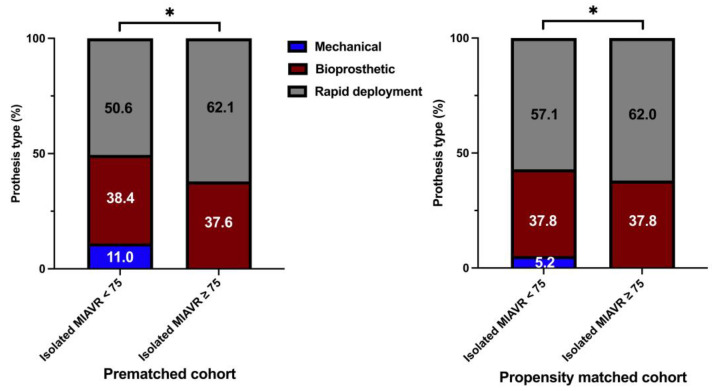
Distribution of aortic prosthesis type implanted within each treatment group. Note: * *p* < 0.05 between groups. As a consequence of the process of mathematical rounding, wherein percentages are rounded up or down to the nearest tenth decimal place, minute deviations of up to 0.1% from the absolute value of 100% can potentially manifest. The stacked column charts for the isolated MIAVR ≥75 cohort in both prematched and matched groups exclude the extremely small proportion (0.3% or 0.2% respectively) of mechanical valve prostheses. A graphical representation in this scale of this small quantity is not meaningful.

**Figure 6 jcm-12-04963-f006:**
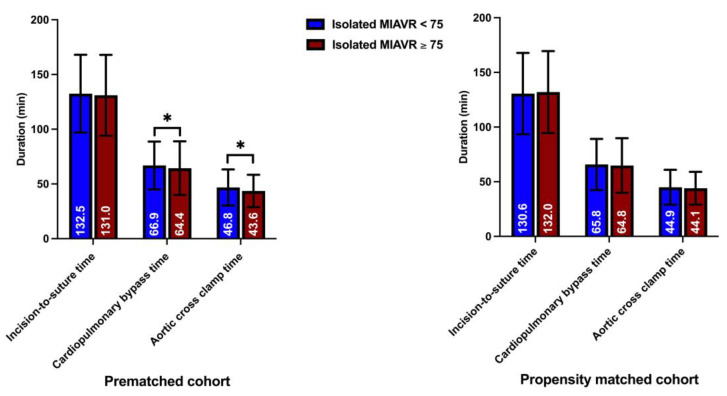
Surgical section times. Note: * *p* < 0.05 between groups.

**Figure 7 jcm-12-04963-f007:**
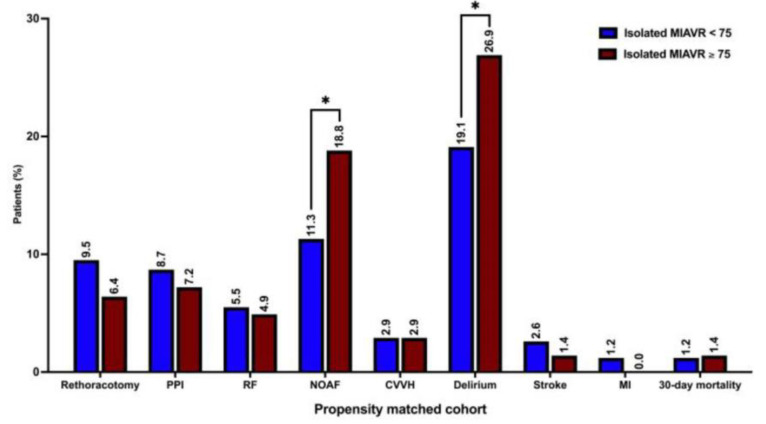
Postoperative outcomes. Abbreviations: PPI, permanent pacemaker implantation; RF, respiratory failure; NOAF, new-onset atrial fibrillation; CVVH, continuous veno-venous hemofiltration; MI, myocardial infarction. Note: * *p* < 0.05 between groups.

**Table 1 jcm-12-04963-t001:** Standardized differences of the variables.

	Before Matching	After Matching
BMI (kg/m^2^)	0.152	0.055
Preoperative LVEF (%)	0.038	0.039
Estimated creatinine clearance (mL/min)	1.023	0.079
NYHA I	0.046	0.023
NYHA II	0.139	0.048
NYHA III	0.150	0.035
NYHA IV	<0.001	0.029
Sex	0.170	0.075
Diabetes (no/yes)	0.167	0.048
Pulmonary hypertension (no/yes)	0.251	0.061
Coronary artery disease (no/yes)	0.380	0.012
Peripheral vascular disease (no/yes)	0.016	0.032
Chronic obstructive pulmonary disease (no/yes)	0.008	0.020

**Table 2 jcm-12-04963-t002:** Baseline characteristics.

	Pre-Matched Cohort			Propensity-Score-Matched Cohort		
	Isolated MIAVR < 75 (*n* = 956)	Isolated MIAVR ≥ 75 (*n* = 383)	*p*	Isolated MIAVR < 75 (*n* = 347)	Isolated MIAVR ≥ 75 (*n* = 347)	*p*
Age (years), mean ± SD	64.2 ± 8.1	77.3 ± 1.7	** *≤0.001 *** **	67.8 ± 5.9	77.2 ± 1.7	** *<0.001 *** **
Sex (male), *n* (%)	623 (65.2)	218 (56.9)	** *≤0.01 ** **	186 (53.6)	199 (57.3)	0.36
Height (cm), mean ± SD	171.2 ± 9.3	168.8 ± 9.3	** *≤0.001 *** **	167.5 ± 9.0	168.7 ± 9.4	0.09
Weight (kg), mean ± SD	83.9 ± 16.7	79.5 ± 13.8	** *≤0.001 *** **	79.4 ± 15.4	79.9 ± 13.9	0.24
BMI (kg/m^2^), mean ± SD	28.6 ± 5.1	27.9 ± 4.2	0.10	28.3 ± 4.9	28.0 ± 4.2	≥0.99
Arterial hypertension, *n* (%)	866 (90.6)	378 (98.7)	** *≤0.001 *** **	332 (95.7)	343 (98.8)	** *0.02 ** **
Diabetes mellitus, *n* (%)	263 (27.5)	135 (35.2)	** *0.02 ** **	119 (34.3)	127 (36.6)	0.91
Dyslipidemia, *n* (%)	565 (59.1)	245 (64.0)	0.11	212 (61.1)	228 (65.7)	0.24
Coronary artery disease, *n* (%)	210 (22.0)	150 (39.2)	** *≤0.001 *** **	132 (38.0)	130 (37.5)	0.77
LVEF (%), mean ± SD	57.3 ± 11.1	57 ± 9.6	0.79	58.1 ± 11.1	57.7 ± 9.6	0.39
COPD, *n* (%)	82 (8.6)	32 (8.4)	≥0.99	33 (9.5)	31 (8.9)	0.90
Pulmonary arterial hypertension, *n* (%)	100 (10.5)	74 (19.3)	** *≤0.001 *** **	55 (15.9)	63 (18.2)	0.45
Renal insufficiency, *n* (%)	118 (12.3)	107 (27.9)	** *≤0.001 *** **	97 (28.0)	83 (23.9)	0.26
Hemodialysis, *n* (%)	10 (1.0)	0 (0.0)	0.07	7 (2.0)	0 (0.0)	** *0.02 ** **
CRCL (mL/min), mean ± SD	91.4 ± 28.0	66.9 ± 19.0	** *≤0.001 *** **	70.5 ± 19.5	68.7 ± 18.8	0.06
PAOD, *n* (%)	37 (3.9)	16 (4.2)	0.54	11 (3.2)	13 (3.7)	0.84
Carotid artery stenosis > 50%, *n* (%)	25 (2.6)	34 (8.9)	** *≤0.001 *** **	132 (38)	130 (37.5)	** *0.02 ** **
TIA, *n* (%)	22 (2.3)	9 (2.3)	≥0.99	10 (2.9)	9 (2.6)	≥0.99
Ischemic stroke, *n* (%)	39 (4.0)	23 (6.0)	0.14	15 (4.3)	22 (6.4)	0.25
Atrial fibrillation, *n* (%)	94 (9.8)	79 (20.7)	** *≤0.001 *** **	47 (13.5)	26 (6.1)	0.22
Pacemaker, *n* (%)	22 (2.3)	21 (5.5)	** *≤0.01 ** **	7 (2.0)	21 (6.1)	** *0.01 ** **
Smoker status, *n* (%)	149 (15.6)	29 (7.6)	** *≤0.001 *** **	37 (10.7)	26 (7.5)	0.19
NYHA class III or IV, *n* (%)	531 (55.5)	241 (62.9)	** *≤0.01 ** **	211 (60.8)	218 (62.8)	0.59
EuroSCORE II (%), mean ± SD	1.38 ± 1.0	2.25 ± 1.3	** *≤0.001 *** **	1.8 ± 1.3	2.2 ± 1.3	** *≤0.001 *** **
STS-PROM Score, mean ± SD	1.1 ± 0.7	1.9 ± 0.9	** *≤0.001 *** **	1.5 ± 0.8	1.9 ± 0.9	** *≤0.001 *** **

Note: Bold and italic values indicate statistical significance: *, *p* ≤ 0.05; **, *p* ≤ 0.01. Abbreviations: MIAVR, minimally invasive aortic valve replacement; SD, standard deviation; BMI, body mass index, LVEF, left ventricular ejection fraction; COPD, chronic obstructive pulmonary disease; CRCL, calculated creatinine clearance according to the Cockcroft–Gault equation; PAOD, peripheral arterial occlusion disease; TIA, transient ischemic attack; NYHA, New York Heart Association; STS-PROM, Society of Thoracic Surgeons predicted risk of mortality.

**Table 3 jcm-12-04963-t003:** Procedural and intraoperative data.

	Pre-Matched Cohort			Propensity-Score-Matched Cohort		
	Isolated MIAVR < 75 (*n* = 956)	Isolated MIAVR ≥ 75 (*n* = 383)	*p*	Isolated MIAVR < 75 (*n* = 347)	Isolated MIAVR ≥ 75 (*n* = 347)	*p*
Surgical access route ^†^						
- UPS, *n* (%)	307 (32.1)	114 (29.8)		110 (31.7)	107 (30.8)	
- RAT, *n* (%)	331 (34.6)	161 (42.0)	** *0.03 ** **	117 (33.7)	142 (40.9)	0.10
- RLA, *n* (%)	318 (33.3)	108 (28.2)		120 (34.6)	98 (28.2)	
Prosthesis size (mm), mean ± SD	23.9 ± 2.0	23.8 ± 2.0	*0.35*	186 (53.6)	199 (57.3)	0.36
STST (min), mean ± SD	171.2 ± 9.3	79.5 ± 13.9	0.44	167.5 ± 9.0	168.7 ± 9.4	0.09
CPBT (min), mean ± SD	66.9 ± 21.8	64.4 ± 24.5	** *0.02 ** **	79.4 ± 15.4	79.9 ± 13.9	0.24
ACCT (min), mean ± SD	46.8 ± 16.5	43.6 ± 14.8	** *≤0.001 *** **	28.3 ± 4.9	28.0 ± 4.2	≥0.99
Prosthesis type ^†^						
- Mechanical, *n* (%)	105 (11.0)	1 (0.3)		18 (5.2)	1 (0.3)	
- Bioprosthetic, *n* (%)	367 (38.4)	144 (37.6)	** ≤*0.001* ** **	131 (37.8)	131 (37.8)	** ≤*0.001* ** **
- RDV, *n* (%)	483 (50.6)	238 (62.1)		198 (57.1)	215 (62.0)	

Note: Bold and italic values indicate statistical significance: *, *p* ≤ 0.05; **, *p* ≤ 0.01; ^†^, see Table 3 for adjusted residuals. As a consequence of the process of mathematical rounding, wherein percentages are rounded up or down to the nearest tenth decimal place, minute deviations of up to 0.1% from the absolute value of 100% can potentially manifest. Abbreviations: MIAVR, minimally invasive aortic valve replacement; UPS, upper partial sternotomy; RAT, right anterolateral thoracotomy; RLA, right lateral thoracotomy; SD, standard deviation; STST, skin-to-skin time; CPBT, cardiopulmonary bypass time; ACCT, aortic cross-clamp time; RDV, rapid deployment bioprosthetic valve.

**Table 4 jcm-12-04963-t004:** Surgical access route and prosthesis type (adjusted residuals for deviations from expected frequencies).

	Pre-Matched Cohort			Propensity-Score-Matched Cohort		
	Isolated MIAVR < 75 (*n* = 956)	Isolated MIAVR ≥ 75 (*n* = 383)	*p*	Isolated MIAVR < 75 (*n* = 347)	Isolated MIAVR ≥ 75 (*n* = 347)	*p*
Surgical access route						
- UPS, asr	0.8	−0.8	>0.05	-	-	>0.05
- RAT, asr	**−2.5**	**2.5**	**≤0.05**	-	-	>0.05
- RLA, asr	1.8	108 (28.2)	>0.05	-	-	>0.05
Prosthesis type						
- Mechanical, asr	**6.6**	**−6.6**	**≤0.05**	**4.0**	**−4.0**	**≤0.05**
- Bioprosthetic, asr	0.3	−0.3	>0.05	0	0	>0.05
- RDV, *n* (%), asr	**−3.8**	**3.8**	**≤0.05**	−1.3	1.3	>0.05

Note: Bold values indicate statistical significance. Abbreviation: MIAVR, minimally invasive aortic valve replacement; asr, adjusted standardized residuals.

**Table 5 jcm-12-04963-t005:** Postoperative morbidity and mortality.

	Pre-Matched Cohort			Propensity-Score-Matched Cohort		
	Isolated MIAVR < 75 (*n* = 956)	Isolated MIAVR ≥ 75 (*n* = 383)	*p*	Isolated MIAVR < 75 (*n* = 347)	Isolated MIAVR ≥ 75 (*n* = 347)	*p*
Ventilation time (hours)						
- ≤12, *n* (%)	869 (91.0)	333 (87.2)		110 (31.7)	107 (30.8)	
- ≤24, *n* (%)	57 (6.0)	33 (8.6)	***0.04* * **	117 (33.7)	142 (40.9)	0.10
- >24, *n* (%)	29 (3.0)	16 (4.2)		120 (34.6)	98 (28.2)	
Respiratory failure ^†^, *n* (%)	41 (4.3)	19 (5.0)	0.56	19 (5.5)	17 (4.9)	0.86
ICU stay (days), mean ± SD	2.1 ± 2.8	2.1 ± 2.1	0.17	2.4 ± 3.6	2.1 ± 1.9	0.57
Hospital stay (days), mean ± SD	9.9 ± 6.4	10.5 ± 5.0	** ≤*0.001* ** **	11.0 ± 8.0	10.5 ± 5.0	0.11
Transfusion (PRBC), mean ± SD	0.7 ± 2.3	0.8 ± 3.6	***0.02* * **	0.9 ± 2.2	0.8 ± 3.7	0.16
AKI, *n* (%)	47 (4.9)	38 (9.9)	** ≤*0.001* ** **	31 (9.0)	32 (9.2)	≥0.99
AKI grade II or III, *n* (%)	36 (3.8)	28 (7.3)	***0.01* * **	26 (7.5)	24 (6.9)	0.83
CVVH, *n* (%)	14 (1.5)	11 (2.9)	0.12	10 (2.9)	10 (2.9)	≥0.99
Conversion to sternotomy, *n* (%)	23 (2.4)	12 (3.1)	0.45	7 (2.0)	11 (3.2)	0.48
Rethoracotomy, *n* (%)	72 (7.5)	27 (7.1)	0.82	33 (9.5)	22 (6.4)	0.16
Impaired wound healing, *n* (%)	81 (8.5)	25 (6.5)	0.26	34 (9.8)	24 (6.9)	0.22
Postoperative delirium, *n* (%)	139 (14.6)	104 (27.2)	** ≤*0.001* ** **	66 (19.1)	93 (26.9)	***0.02* * **
Ischemic stroke, *n*	21 (2.2)	8 (2.1)	≥0.99	9 (2.6)	5 (1.4)	0.06
TIA, *n* (%)	9 (0.9)	3 (0.8)	≥0.99	5 (1.4)	3 (0.9)	0.73
PPM implantation, *n* (%)	61 (6.4)	27 (7.1)	0.33	30 (8.7)	25 (7.2)	0.68
NOAF, *n* (%)	117 (12.3)	70 (18.3)	** ≤*0.01* * **	39 (11.3)	65 (18.8)	** ≤*0.01* * **
Myocardial infarction, *n* (%)	10 (1.0)	0 (0.0)	0.33	4 (1.2)	0 (0.0)	0.12
30-day mortality, *n* (%)	14 (1.5)	13 (3.4)	0.82	4 (1.2)	5 (1.4)	0.90

Note: Bold and italic values indicate statistical significance: *, *p* ≤ 0.05; **, *p* ≤ 0.01; ^†^, defined as primary postoperative ventilation time ≥72 h, reintubation and tracheotomy. Abbreviations: MIAVR, minimally invasive aortic valve replacement; AKI, acute kidney injury; CVVH, consecutive renal failure needing continuous veno-venous hemofiltration; ICU, intensive care unit; PRBC, packed red blood cells; TIA, transient ischemic attack; NOAF, new-onset atrial fibrillation; PPM, permanent pacemaker.

**Table 6 jcm-12-04963-t006:** Echocardiographic data (on admission and at discharge).

	Pre-Matched Cohort			Propensity-Score-Matched Cohort		
	Isolated MIAVR < 75 (*n* = 956)	Isolated MIAVR ≥ 75 (*n* = 383)	*p*	Isolated MIAVR < 75 (*n* = 347)	Isolated MIAVR ≥ 75 (*n* = 347)	*p*
Preoperative AVA (cm^2^), mean ± SD	0.7 ± 0.2	0.7 ± 0.2	0.34	0.7 ± 0.2	0.7 ± 0.1	0.89
Preoperative P_max_ (mmHg), mean ± SD	78.4 ± 26.1	75.0 (21.8)	***0.01* * **	77.3 ± 32.3	73.8 ± 20.4	0.15
Preoperative P_mean_ (mmHg), mean ± SD	47.6 ± 14.6	46.0 ± 15.0	***0.01* * **	46.7 ± 14.9	45.2 ± 14.1	0.12
Postoperative P_max_ (mmHg), mean ± SD	24.8 ± 8.1	24.5 ± 7.9	0.70	11.0 ± 8.0	10.5 ± 5.0	0.11
Postoperative P_mean_ (mmHg), mean ± SD	14.1 ± 4.6	14.0 ± 4.6	0.76	14.0 ± 4.7	14.1 ± 4.7	0.98
Paravalvular AR, *n* (%)	31 (3.3)	10 (2.6)	0.55	11 (3.2)	10 (2.9)	≥0.99
Paravalvular AR ≥ II, *n* (%)	13 (1.4)	4 (1.1)	0.30	2 (0.6)	4 (1.2)	0.68

Note: Bold and italic values indicate statistical significance: *, *p* ≤ 0.05. Abbreviations: MIAVR, minimally invasive aortic valve replacement; AVA, aortic valve area; P_max_, peak aortic valve gradient; P_mean_, mean aortic valve gradient; AR, aortic regurgitation.

## Data Availability

The data presented in this study are available upon request from the corresponding author. The data are not publicly available due to ethical regulations.

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
