# Peer review of "Minimally Invasive Isolated Aortic Valve Replacement in a Potential TAVI Cohort of Patients Aged ≥ 75 Years: A Propensity-Matched Analysis"

_jcm, 2023, doi:10.3390/jcm12154963_

Round 1

Reviewer 1 Report

In this study, Dr. Taghizadeh-Waghefi and colleagues present the results of a retrospective analysis of all patients undergoing aortic valve replacement (AVR) with an approach other than full sternotomy over a 9-year period at a single institution (by a group of multiple surgeons, using a variety of surgical approaches). The study compared outcomes between patients > 75 years old and those ≥ 75. The primary outcomes compared were 30-day mortality rates and major adverse cardiac and cerebrovascular events (MACCE). The apparent impetus for this study was the recently published European valve guidelines, which suggested that an age cut-off of 75 be used in decisions for approach to AVR.

The study group included 1,339 patients. In addition to comparing results between age groups, the authors used propensity score matching to compare results within 347 pairs, matched based on baseline characteristics. The authors report no significant differences in 30-day mortality or MACE based on this age cut-off, in either the un-adjusted comparison or in the propensity-matched comparison. There were differences between groups in time to extubation and transfusion rates, and in the rates of postoperative acute kidney injury, delirium and atrial fibrillation. However, only atrial fibrillation and delirium rates were different between groups after propensity matching.

This study is well-conceived and executed. The manuscript is concise and extremely well-written. The authors make a strong argument that based both on existing literature and their analysis, the guideline recommendation for an age cut-off to dictate approach to AVR is not rooted in evidence.

The study is limited in its retrospective design and only short-term follow-up, as noted by the authors. It also presents a heterogeneous group of surgical approaches (including partial sternotomy and several variations of mini-thoracotomy). I feel this article is both interesting and timely. I do have several questions/comments for the authors:

11)   Can the authors comment on why they chose to only include/compare patients undergoing “minimally invasive” isolated AVR? Do they really feel that the results would have been different had they analyzed the entire group of patients undergoing isolated surgical AVR? Regardless, the outcomes of patients undergoing full sternotomy would be of interest (since this remains the primary approach to surgical AVR in most centers).

22)   In the introduction, the authors note that this study sought to compare outcomes between patients “aged ≥ 75 with a life expectancy > 5 years” to younger patients. However, I see nothing in the study methodology (or study flow diagram) where elderly patients with a lesser predicted life expectancy were excluded.

33)   In the U.S., guidelines suggest an age cut-off of 65 years in guiding the approach to AVR. Similar to European guidelines, this recommendation appears to be based on minimal evidence. In practice, patient demands are largely driving a surge in trans-catheter AVR for younger and low-risk AS patients in the U.S., regardless of the absence of longer term data supporting this practice. Can the authors comment on why they feel the suggested age cut-off is so disparate between the U.S. and European guidelines?

Author Response

Dear Reviewer,

Thank you for your constructive comments. You will find our point-by-point response in the Word attachment file. We hope that we have given them due consideration and that the manuscript could be significantly improved with your help.

Yours sincerely
Ali Waghefi on behalf of the authors

Reviewer 2 Report

Thank you for the opportunity to review this interesting paper that features the outcomes of patients , according to age over or under 75 years, who underwent minimal invasive aortic valve replacement in a single high-volume cardiac surgery center.

I congratulate the authors for this well constructed, well written study. However, I do not fully agree with the authors' intention and interpretation of the results. This study presents the results of the MIAVR in patients according to age over or under 75 years. Indeed, there is no difference in mortality in this cohort, and the indication for TAVI versus MIAVR in patients over 75 years of age should be discussed by a heart team and according to the patient's wishes. 

However, in the absence of a direct comparison between MIAVR and TAVI patients over 75 years old, I do not understand how this study allows to question the current recommendations on the indication of TAVI in patients over 75 years old. Furthermore, the authors could also provide outcomes of patients over 75 years old who have undergone AVR by complete sternotomy in order to have a comparison between full sternotomy SAVR, MIAVR and TAVi in the over 75 years old. 

Although we might agree that the  the cut off age determined in the guidelines can be discussed, this study does not provide direct evidence that MIAVR is safer than TAVI.

Furthermore, these MIAVR results in an expert centre like this with a high volume of patients cannot be extrapolated to all patients.

Minors comments: 

- Although it is a retrospective study, was this study approved by an ethical committee? 

- Table 2, I think there is a mistake in ICU length of stay in matched cohort, in hours?

Author Response

Dear Reviewer,

Thank you very much for your constructive comments. You will find our point-by-point response in the Word attachment. We hope that we have taken them into account and that with your help the manuscript can be significantly improved.

Yours sincerely.
Ali Waghefi on behalf of the authors

Round 2

Reviewer 2 Report

The authors have responded appropriately to my comments and I feel that the paper has improved with the changes made.

Author Response

Dear Reviewer 2,

we would like to thank you for your positive review of our papers.

Best regards,

Ali Taghizadeh-Waghefi on behalf of all authors